# What are the implications of using individual and combined sources of routinely collected data to identify and characterise incident site-specific cancers? a concordance and validation study using linked English electronic health records data

Helen Strongman ,[1] Rachael Williams,[2] Krishnan Bhaskaran[1]

For numbered affiliations see end of article.

**Correspondence to**
Dr Helen Strongman;
helen.strongman@lshtm.ac.uk

## ABSTRACT

**Objectives** To describe the benefits and limitations of using individual and combinations of linked English electronic health data to identify incident cancers.

**Design and setting** Our descriptive study uses linked English Clinical Practice Research Datalink primary care; cancer registration; hospitalisation and death registration data.

**Participants and measures** We implemented case definitions to identify first site-specific cancers at the 20 most common sites, based on the first ever cancer diagnosis recorded in each individual or commonly used combination of data sources between 2000 and 2014. We calculated positive predictive values and sensitivities of each definition, compared with a gold standard algorithm that used information from all linked data sets to identify first cancers. We described completeness of grade and stage information in the cancer registration data set.

**Results** 165 953 gold standard cancers were identified. Positive predictive values of all case definitions were ≥80% and ≥94% for the four most common cancers (breast, lung, colorectal and prostate). Sensitivity for case definitions that used cancer registration alone or in combination was ≥92% for the four most common cancers and ≥80% across all cancer sites except bladder cancer (65% using cancer registration alone). For case definitions using linked primary care, hospitalisation and death registration data, sensitivity was ≥89% for the four most common cancers, and ≥80% for all cancer sites except kidney (69%), oral cavity (76%) and ovarian cancer (78%). When primary care or hospitalisation data were used alone, sensitivities were generally lower and diagnosis dates were delayed. Completeness of staging data in cancer registration data was high from 2012 (minimum 76.0% in 2012 and 86.4% in 2014 for the four most common cancers).

**Conclusions** Ascertainment of incident cancers was good when using cancer registration data alone or in combination with other data sets, and for the majority

## Strengths and limitations of this study

► This is the first study to present comprehensive information on the implications of using different individual and combinations of linked electronic health data sources in England to identify cases of the 20 most common incident cancers.

► Using a gold standard algorithm that combined all available data from multiple sources as a comparator, we were able to estimate both positive predictive values and sensitivity values for a range of pragmatic case definitions.

► We described similarities and differences in values between age groups, sexes and calendar years, the impact of choice of source(s) on diagnosis dates and mortality rates and completeness of stage and grade in cancer registration data.

► A key limitation was that our gold standard algorithm is not validated and may be affected by differences in clinical diagnosis and coding of invasive cancers between data sources.

of cancers when using a combination of primary care, hospitalisation and death registration data.

## INTRODUCTION

The Clinical Practice Research Datalink (CPRD) provides de-identified primary care data linked to additional secondary health data sources, under a well-governed framework.[1] Use of linked data helps researchers to answer more epidemiological questions and increase study quality through improved exposure, outcome and covariate classification.[2] In the field of cancer epidemiology, CPRD primary care data linked to Hospital Episode Statistics Admitted Patient Care



data (HES APC), Office of National Statistics (ONS) mortality and National Cancer Registration and Analysis Service (NCRAS) cancer registration data are used to analyse factors contributing to the risk of cancer and the consequences of cancer and its treatment. Use of linked data reduces the sample to the common source population and data coverage period for each included data set, and has cost and logistical implications, which are greatest for NCRAS data. Research teams therefore commonly choose not to use all available linked data.[3] Cancer epidemiology studies can also be conducted using NCRAS and HES APC data provided by National Health Service (NHS) Digital and Public Health England (PHE), without linkage to CPRD primary care data.[4] This provides national coverage at the expense of the detailed health data that are available in primary care records.

Validation studies assessing concordance between CPRD GOLD, HES APC and NCRAS data have estimated high positive predictive values (PPVs) for CPRD GOLD data and varying proportions of registered cancers that are not captured in CPRD GOLD and HES APC.[5–8] The most up-to-date analysis by Arhi *et al* included the five most common cancers and all papers focussed on concordance between CPRD GOLD only and NCRAS; existing evidence therefore does not provide a complete assessment of the benefits and limitations of using different combinations of data sources within the context of practical study designs. National data are available describing completeness of data fields within the cancer registry data in each collection year[9] and over time for all cancers combined[4]; missingness for individual years has been associated with age, comorbidities and Clinical Commissioning Groups.[10 11]

We aim to describe and compare the benefits and limitations of using different combinations of linked CPRD primary care data, HES APC, ONS mortality and NCRAS cancer registration data, for conducting cancer epidemiology studies. Our analyses focus on incident cancer ascertainment as it is a common and important outcome in cancer epidemiology, and it is more difficult to distinguish between secondary, recurrent and primary cancers at a second site in these data sets. We have compared definitions of the 20 most common cancers based on the first ever cancer recorded in individual or combinations of data sets with a gold standard definition comparing information from all four data sets. We also describe the availability of stage, grade and treatment variables over time in the cancer registration data for the CPRD linked cohort. This reflects real-life study design and will help researchers to decide which combination of data sources to use for future studies.

## METHODS
### Study design and setting
We completed a concordance study using linked[2] English CPRD GOLD, HES APC, ONS mortality and NCRAS data. CPRD GOLD data were extracted from the January 2017 monthly release and the 13th update to CPRD's linked data. The study period was from 1 January 2000 to 31 December 2014, with 31 December 2014 matching the end of the NCRAS coverage period.

The CPRD GOLD database includes de-identified records from participating general practices in the UK who use Vision software.[1] General practice staff can record cancer diagnoses using Read codes or in free text comments boxes, though the latter are not collected by CPRD. Diagnoses will typically be entered during/following a consultation or from written information that is returned to the practice from secondary care. CPRD GOLD data are linked to HES APC, ONS mortality and NCRAS through a trusted third party for English practices that have agreed to participate in the linkage programme.[2] HES APC data are collected by NHS Digital to co-ordinate clinical care in England and calculate hospital payments.[12] Admissions for and related to cancer diagnoses are recorded using International Classification of Diseases, version 10 (ICD-10) codes. National cancer registration data are collected by NCRAS which is part of PHE in accordance with the Cancer Outcomes and Services Data set[13] which has been the national standard for reporting of cancer in England since January 2013. Data include ICD-10 codes to identify the cancer site and more detailed information such as stage and grade. ONS mortality data includes dates and causes of deaths registered in England, recorded using ICD-10 codes.

### Participants, exposures and outcomes
Our underlying study population included male and female patients registered in CPRD GOLD practices who were eligible for linkage to HES APC, NCRAS and ONS mortality data and had at least 365 days of follow-up between 1 January 1999 and 31 December 2014. Start of follow-up was defined as the latest of the current registration date within the practice and the CPRD estimated start of continuous data collection for the practice (up-to-standard date). End of follow-up was determined as the date the patient left the practice, ONS mortality date of death or practice last collection date.

### Identification and classification of cancer codes
We used code lists to classify cancer records in each of CPRD GOLD, HES APC and ONS mortality data as one of the 20 most common sites, other specified cancers, history of cancer, secondary cancers, benign tumours, administrative cancer codes, unspecified and incompletely specified cancer codes (https://doi.org/10.17037/data. 00001519). Incompletely specified cancer codes could be mapped to >1 cancer site (eg, ICD-10 code C68.9 "Malignant neoplasms of urinary organ unspecified" was considered consistent with both bladder and kidney cancer). For NCRAS, we accessed coded records for the 20 most common cancers. We included cancers recorded in the clinical or referral file for CPRD GOLD, cancers recorded in any diagnosis field for HES APC and the underlying or

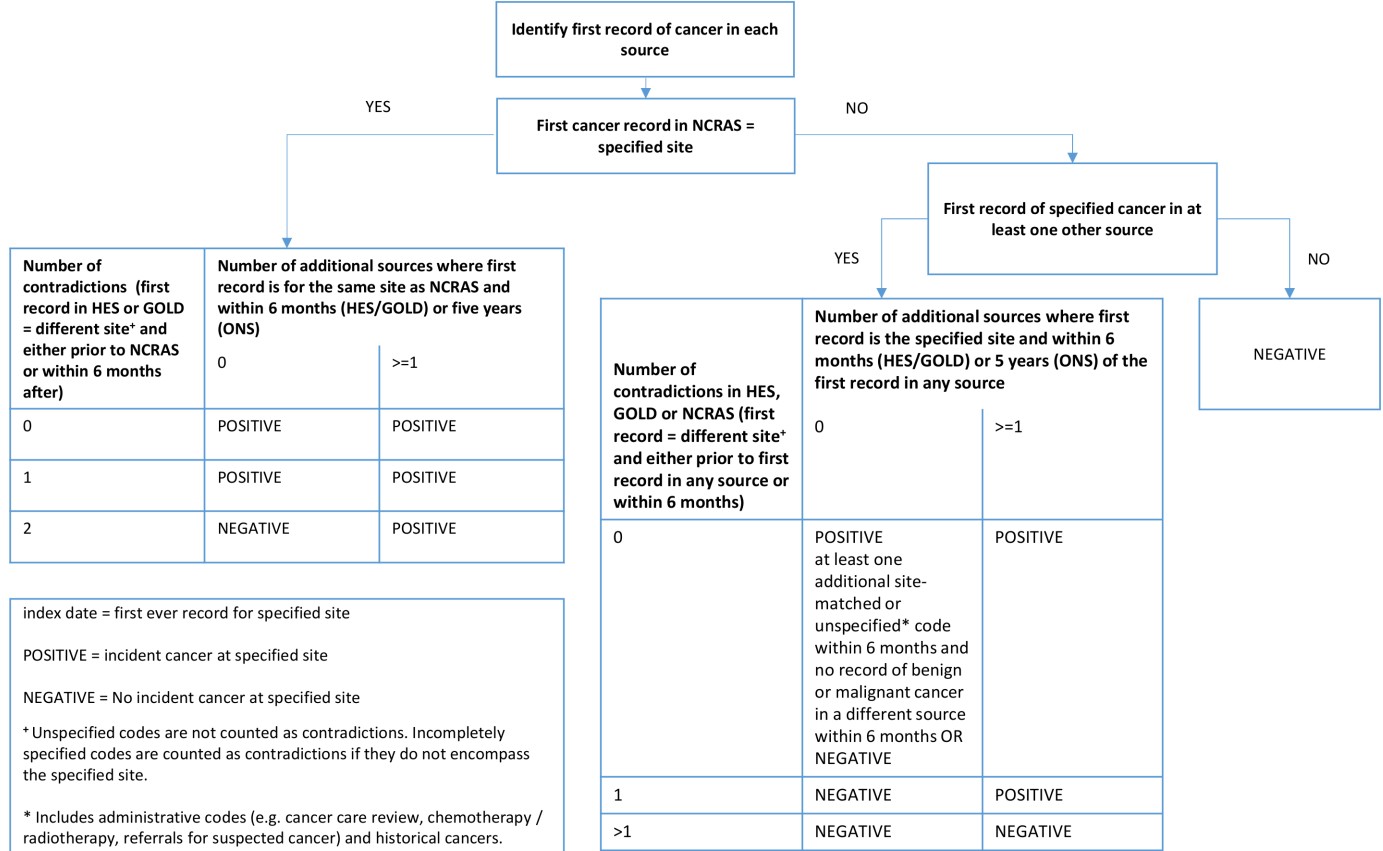

**Figure 1** Gold standard algorithm to identify incident site-specific cancers using all data sources. HES, Hospital Episode Statistics; NCRAS, National Cancer Registration and Analysis Service; ONS, Office of National Statistics.

most immediate cancer cause of death in ONS mortality data.

## Cancer case definitions based on individual sources and combinations of sources

We developed alternative cancer case definitions mirroring those commonly used in epidemiology studies, based on identifying the first malignant cancer (excluding administrative codes and benign tumours) recorded in various combinations of data sources (NCRAS alone; NCRAS and HES APC; all sources; CPRD GOLD, HES APC and ONS mortality; CPRD GOLD alone, HES APC alone). Multiple malignant cancers recorded on the index date in CPRD GOLD or HES APC were reclassified as multiple-site cancer and were not considered as individual-site cancer records for positive predictive value and sensitivity calculations; multiple codes recorded in different sources on the same date were reclassified as the site identified in the NCRAS data if available and as multiple-site cancer if not. For each case definition, we only examined the first malignant cancer per individual where this occurred within the study period and at least 1 year after the start of follow-up.

## Gold standard cancer case definition

We developed a gold standard algorithm that classifies incident records of the 20 most common cancers by comparing the first malignant cancer identified in each individual source (figure 1). Cancers recorded in NCRAS alone with no contradictions (ie, records for first cancers at different sites) were considered true cases whereas cancers recorded in HES APC alone or GOLD alone required internal confirmation within that source in the form of another code for cancer consistent with the same site (or with site unspecified) within 6 months and no contradictory codes (eg, for cancers at other sites) in this period. Where cancer records were present in >1 data source, we considered a site-specific cancer to be a true case (a) if it was recorded as the first cancer in NCRAS and the total number of data sources with records for cancer at that site was equal to or greater than the number of data sources with contradictory records (ie, records for first cancers at different sites); or (b) where the cancer was not present in NCRAS, if there were more data sources in total with records for cancer at that site than data sources with contradictory records.

We used NCRAS data to identify stage, grade and treatment where available in the cancer registry only cohort. Binary surgery, chemotherapy and radiotherapy variables were derived using individual records of treatment from the first year after diagnosis.

## Statistical analysis

For each cancer site and each individual or combined data source, we combined our applied study definitions

with our gold standard definition to classify each applied study definition as a true positive, false positive or false negative record.

We used these categories to calculate sensitivity and positive predictive value overall and stratified by age categories (<60, 60 to 79 and 80+), calendar year and sex. We calculated differences in diagnosis dates for true positives by subtracting the gold standard index date from the index date for each source and combination of sources.

We used Kaplan-Meier methods to describe mortality over time for cancers identified using each definition. The ONS mortality death date was used for these analyses.

We used the NCRAS only definition to calculate proportions of patients with complete stage and grade and recorded cancer treatment modalities over time.

### Patient public involvement

Patients and the public were not involved in conceiving, designing or conducting this study and will not be consulted regarding the dissemination of study results.

### RESULTS

Of 14 747 047 research quality patients in the CPRD GOLD January 2017 build, 8 893 326 were eligible for linkage to HES, ONS mortality and NCRAS data in set 13; 237 were excluded due to unknown sex. Of the remainder, 6 791 074 and had at least 1 year of follow-up between 1 January 1999 and 31 December 2014 and were included in the study population. Using the gold standard algorithm, 165 953 incident cases of cancer were identified. The number of patients identified with each cancer is presented in online supplementary appendix table 1. Half (50.0%, n=82 899) of these patients were male; 24.4% (40 470) aged 0 to 59, 54.1% (89 720) aged 60 to 79 and 21.6% (35 763) aged 80 or older.

Figure 2 presents PPVs for each case definition, comparing the first recorded cancer in each combination of data sources with the gold standard algorithm. When using NCRAS data alone, 91.0% to 99.5% of cancers were confirmed by the algorithm; for 19 out of 20 cancer sites, the NCRAS-only case definition gave the highest PPV. Case definitions using data sources not including NCRAS generally had lower PPVs, ranging from 79.6% to 97.3% for individual cancer sites. For the four most common cancers (breast, lung, colorectal and prostate), PPVs were at least 94% for all case definitions. Minimal differences in PPVs were observed between age groups, years and sexes (online supplementary appendix figures 1–3).

Figure 3 presents sensitivity values for each case definition. Sensitivity was generally higher for the case definitions that included NCRAS data (ranging from 80.9% to 98.7% for individual cancer sites except bladder cancer identified using NCRAS data alone (64.8%), and ≥92% for the four most common cancers (breast, lung, colorectal and prostate)). Sensitivity was also generally high for definitions using a combination of CPRD GOLD, HES APC and ONS mortality data (ranging from 69.2%

to 96.3%, ≥89% for the four most common cancers). Sensitivity was lower for case definitions that used CPRD GOLD alone (range 31.5% to 89.3% for individual cancer sites) or HES APC alone (range 55.9% to 92.6%). Sensitivity values for CPRD GOLD alone and HES APC alone increased slightly in younger patients and more recent years; no differences were observed between men and women (online supplementary appendix figures 4–6). Post-hoc analysis suggested that the low sensitivity of CPRD GOLD only definitions for kidney cancer (sensitivity 31.5%, n false negatives 2869) was driven by missing (n=1136, 39.6%) or incompletely specified urinary organ cancer codes (n=1108, 38.6%) in CPRD GOLD rather than contradictory information about the first cancer record (n=625, 21.8%). These incompletely specified codes are less likely to be used for bladder cancers (n=85) than kidney cancers (n=1108). Bladder cancers that were not recorded in NCRAS data (n=3445) were commonly recorded in both HES APC and CPRD GOLD (n=2228, 64.7%) or in HES APC only with a subsequent unspecified or bladder cancer record in HES APC within 6 months (n=995, 28.9%).

Table 1 describes the number of days (median IQR and 5th/95th percentile) lag between the date of incident cancers from the gold standard definition and the date of cancer arising from each case definition (ie, the first record within the specific combinations of data sources used). Case definitions using NCRAS alone and combinations of ≥2 data sources captured cancers close to the gold standard date (median lag ≤7 days for all cancer sites), whereas median lags were generally longer for the case definitions using CPRD GOLD alone and HES APC alone.

Figure 4 describes mortality over time following incident cancer diagnoses ascertained from each case definition. Minimal differences in mortality were observed between cancers identified from different case definitions. Where variability was observed, cancers identified using CPRD GOLD only had the lowest mortality rates (eg, kidney cancer) and cancers identified using HES APC only or NCRAS only had higher mortality rates (eg, prostate cancer and bladder cancer, respectively).

Figure 5 describes completeness of grade and stage for cancers identified using NCRAS only. Recording of grade was highly variable between cancers with gradual increases in completeness over time. Completeness of staging information was low in earlier calendar years but improved substantially from around 2012 especially for the four most common cancers (minimum 76.0% in 2012 and 86.4% in 2014). Post-hoc logistic regression models adjusted for year and cancer site indicated that completeness of stage and grade were associated with each other and these variables were least complete in patients aged >=80; stage data was more complete for higher grade tumours whereas grade data was more complete for lower stage tumours (online supplementary appendix figure 7).

Online supplementary appendix figure 8 describes recording of treatment modalities identified using NCRAS

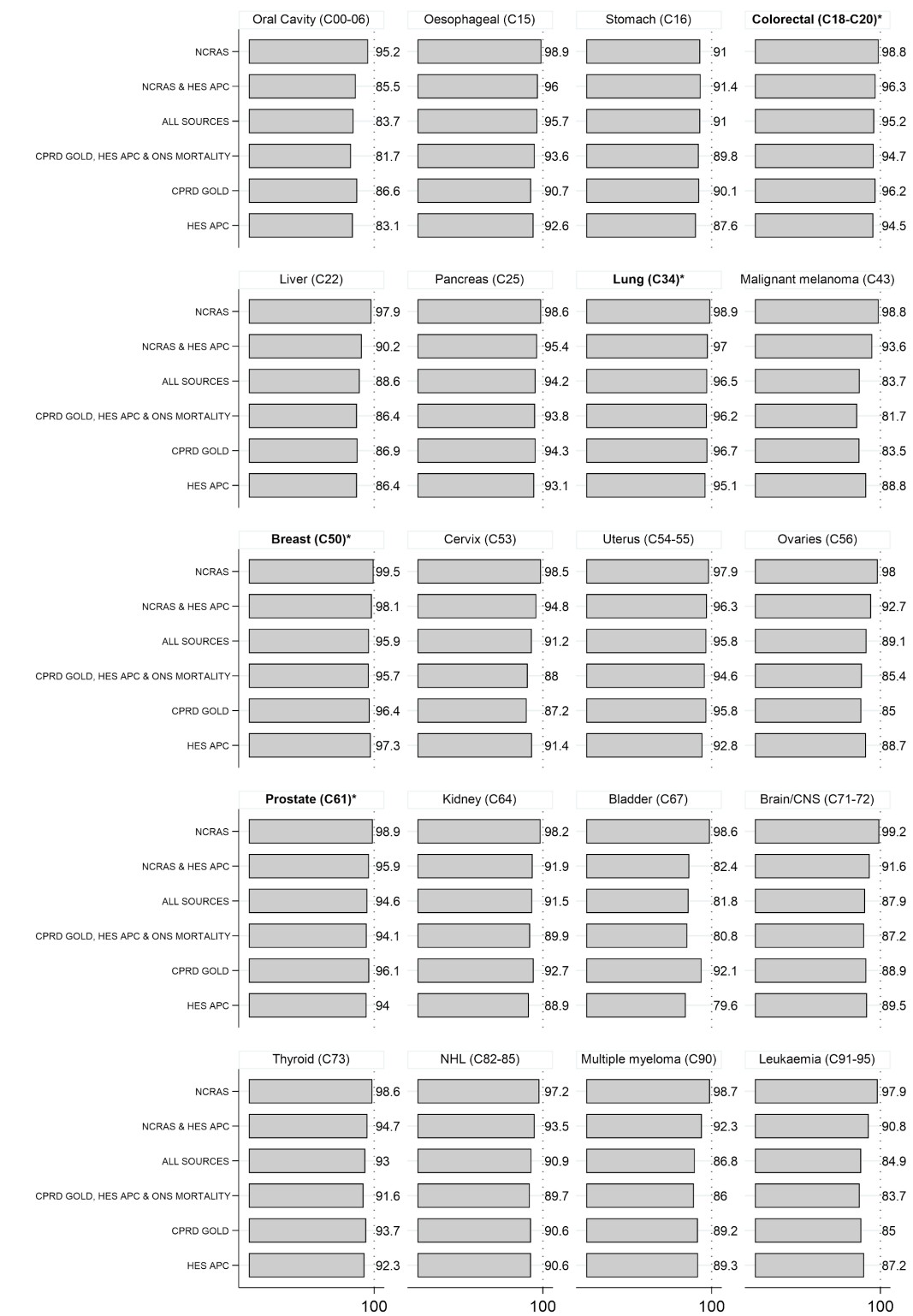

**Figure 2** Positive Predictive Value of cancer diagnoses for each combination of sources when compared with the main gold standard algorithm. Percentage of incident cancers defined using the first ever record in each combination of sources confirmed by a gold standard algorithm that considers confirmatory and contradictory data from each source. Cancer sites are ordered according to corresponding codes from the International Classification of Diseases, version 10 (ICD-10). *Four most common cancer sites. CNS, central nervous system; NHL, Non-Hodgkin's lymphoma; CPRD, Clinical Practice Research Datalink; HES APC, Hospital Episode Statistics Admitted Patient Care data; NCRAS, National Cancer Registration and Analysis Service; ONS, Office of National Statistics.

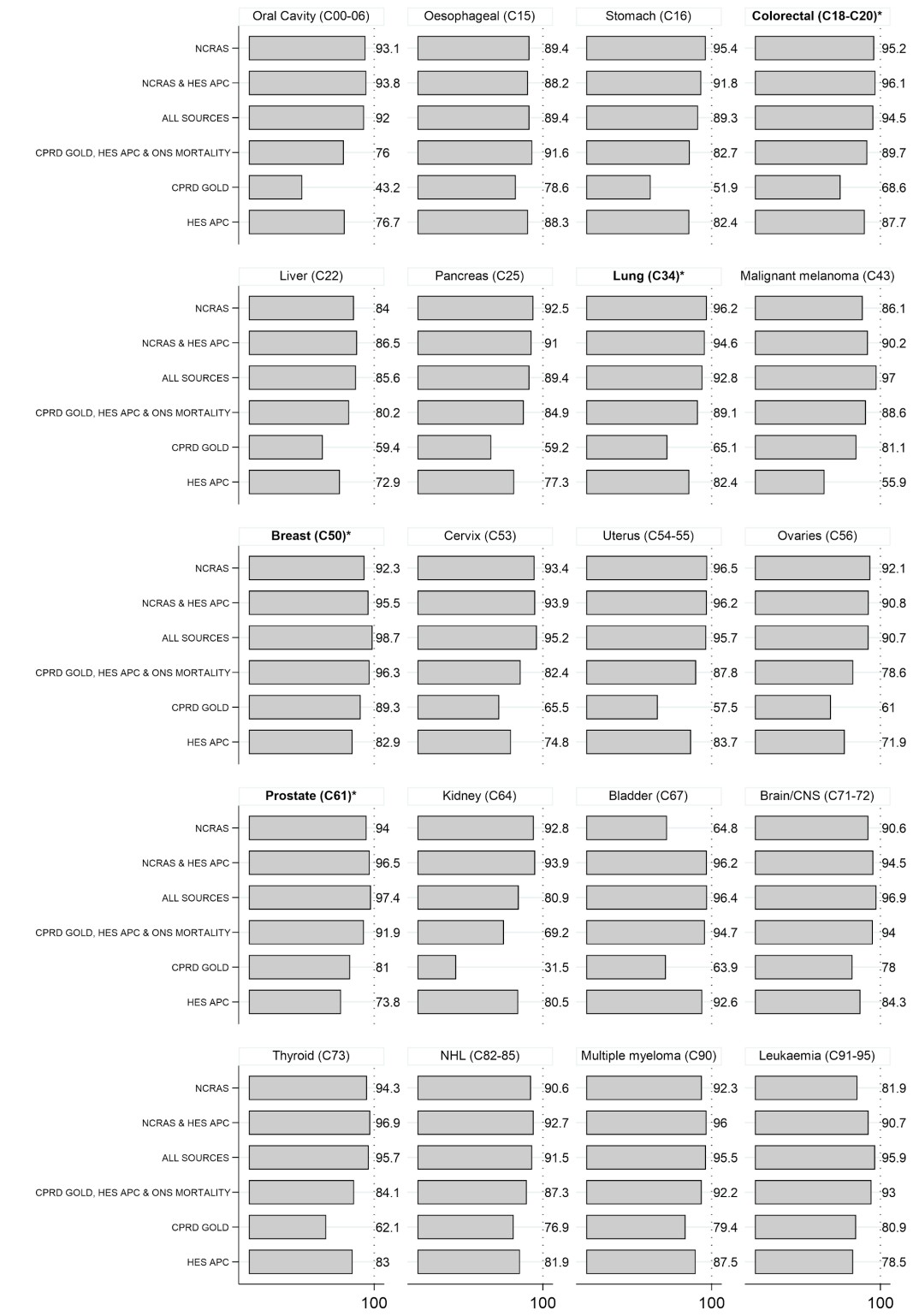

**Figure 3** Sensitivity of cancer diagnoses for each combination of sources when compared with the main gold standard algorithm. Percentage of incident cancers identified using the main gold standard algorithm that considers confirmatory and contradictory data from each source that are identified using the first ever record in each combination of sources. Cancer sites are ordered according to corresponding codes from the International Classification of Diseases, version 10 (ICD-10). *Four most common cancer sites. CNS, central nervous system; NHL, Non-Hodgkin's lymphoma; CPRD, Clinical Practice Research Datalink; HES APC, Hospital Episode Statistics Admitted Patient Care data; NCRAS, National Cancer Registration and Analysis Service; ONS, Office of National Statistics.

**Table 1** Time in days from main gold standard diagnosis date to first ever record in each combination of sources

| Cancer | NCRAS | | NCRAS and HES APC | | CPRD GOLD, HES APC and ONS mortality | | CPRD GOLD | | HES APC | |
|---|---|---|---|---|---|---|---|---|---|---|
| | Median (IQR) | 5th–95th percentile | Median (IQR) | 5th–95th percentile | Median (IQR) | 5th–95th percentile | Median (IQR) | 5th–95th percentile | Median (IQR) | 5th–95th percentile |
| Oral cavity (C00–06) | 0 (0 to 0) | 0–20 | 0 (0 to 0) | 0–12 | 0 (0 to 17) | 0–57 | 11 (0 to 25) | 0–80 | 12 (0 to 39) | 0–91 |
| Oesophageal (C15) | 0 (0 to 1) | 0–30 | 0 (0 to 0) | 0–6 | 0 (0 to 0) | 0–30 | 7 (0 to 18) | 0–59 | 0 (0 to 6) | 0–85 |
| Stomach (C16) | 0 (0 to 2) | 0–28 | 0 (0 to 0) | 0–0 | 0 (0 to 0) | 0–37 | 10 (1 to 22) | 0–64 | 0 (0 to 0) | 0–64 |
| Colorectal (C18–C20)* | 0 (0 to 3) | 0–41 | 0 (0 to 0) | 0–19 | 0 (0 to 0) | 0–36 | 7 (0 to 21) | 0–70 | 0 (0 to 15) | 0–90 |
| Liver (C22) | 0 (0 to 7) | 0–87 | 0 (0 to 0) | 0–51 | 0 (0 to 2) | 0–72 | 9 (0 to 29) | 0–113 | 0 (0 to 32) | 0–170 |
| Pancreas (C25) | 0 (0 to 8) | 0–56 | 0 (0 to 0) | 0–23 | 0 (0 to 0) | 0–52 | 8 (0 to 22) | 0–76 | 0 (0 to 8) | 0–101 |
| Lung (C34)* | 0 (0 to 5) | 0–42 | 0 (0 to 0) | 0–20 | 0 (0 to 4) | 0–56 | 10 (0 to 22) | 0–85 | 0 (0 to 19) | 0–190 |
| Malignant melanoma (C43) | 0 (0 to 0) | 0–23 | 0 (0 to 0) | 0–29 | 0 (0 to 21) | 0–64 | 11 (0 to 25) | 0–73 | 31 (0 to 61) | 0–240 |
| Breast (C50)* | 0 (0 to 0) | 0–26 | 0 (0 to 0) | 0–27 | 7 (0 to 14) | 0–37 | 7 (0 to 14) | 0–48 | 27 (16 to 41) | 0–365 |
| Cervix (C53) | 0 (0 to 0) | 0–17 | 0 (0 to 0) | 0–3 | 3 (0 to 20) | 0–74 | 13 (4 to 27) | 0–79 | 17 (0 to 48) | 0–113 |
| Uterus (C54–55) | 0 (0 to 0) | 0–19 | 0 (0 to 0) | 0–4 | 0 (0 to 19) | 0–55 | 14 (7 to 27) | 0–69 | 8 (0 to 41) | 0–89 |
| Ovaries (C56) | 0 (0 to 3) | 0–33 | 0 (0 to 0) | 0–21 | 0 (0 to 0) | 0–41 | 10 (0 to 24) | 0–95 | 0 (0 to 14) | 0–96 |
| Prostate (C61)* | 0 (0 to 0) | 0–68 | 0 (0 to 0) | 0–82 | 2 (0 to 22) | 0–154 | 15 (3 to 29) | 0–112 | 65 (0 to 423) | 0–2113 |
| Kidney (C64) | 0 (0 to 5) | 0–66 | 0 (0 to 0) | 0–36 | 0 (0 to 0) | –24–78 | 0 (0 to 22) | 0–112 | 0 (0 to 20) | 0–250 |
| Bladder (C67) | 1 (0 to 15) | 0–222 | 0 (0 to 0) | 0–31 | 0 (0 to 0) | 0–29 | 7 (0 to 30) | 0–166 | 0 (0 to 0) | 0–99 |
| Brain/CNS (C71–72) | 1 (0 to 8) | 0–63 | 0 (0 to 0) | 0–31 | 0 (0 to 0) | 0–32 | 8 (0 to 20) | 0–68 | 0 (0 to 1) | 0–166 |
| Thyroid (C73) | 0 (0 to 0) | 0–28 | 0 (0 to 0) | 0–20 | 0 (0 to 25) | 0–87 | 22 (3 to 42) | 0–127 | 1 (0 to 58) | 0–154 |
| Non-Hodgkin's lymphoma (C82–85) | 0 (0 to 3) | 0–43 | 0 (0 to 0) | 0–33 | 0 (0 to 12) | 0–61 | 16 (4 to 32) | 0–118 | 0 (0 to 31) | 0–551 |
| Multiple myeloma (C90) | 0 (0 to 8) | 0–235 | 0 (0 to 0) | 0–80 | 0 (0 to 1) | 0–75 | 10 (0 to 28) | 0–148 | 0 (0 to 41) | 0–714 |
| Leukaemia (C91–95) | 0 (0 to 7) | 0–909 | 0 (0 to 1) | 0–1038 | 0 (0 to 0) | 0–89 | 1 (0 to 20) | 0–140 | 0 (0 to 180) | 0–1811 |

Number of days between main gold standard diagnosis date and applied definitions. Cancer sites are ordered according to corresponding codes from the International Classification of Diseases, version 10 (ICD-10).

*Four most common cancer sites. All sources definition not shown as diagnosis date is the same as the gold standard definition by default.

CNS, central nervous system; CPRD, Clinical Practice Research Datalink; HES APC, Hospital Episode Statistics Admitted Patient Care data; NCRAS, National Cancer Registration and Analysis Service cancerregistration data; ONS, Office for National Statistics.

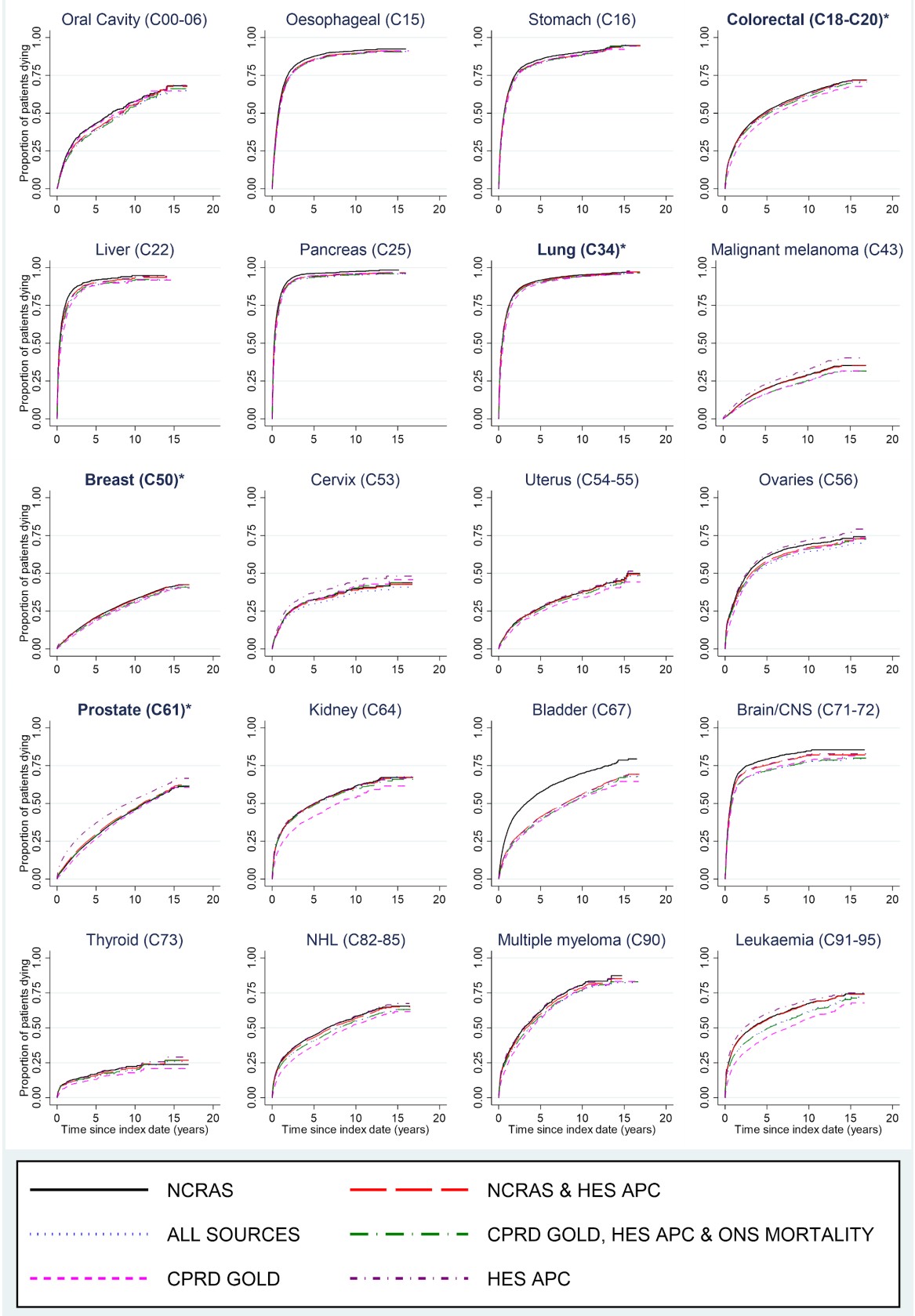

**Figure 4** Mortality following first ever record of cancer in each combination of sources. Cancer sites are ordered according to corresponding codes from the International Classification of Diseases, version 10 (ICD-10). *Four most common cancer sites. CNS, central nervous system; CPRD, Clinical Practice Research Datalink; HES APC, Hospital Episode Statistics Admitted Patient Care data; NCRAS, National Cancer Registration and Analysis Service; NHL, Non-Hodgkin's lymphoma; ONS, Office of National Statistics.

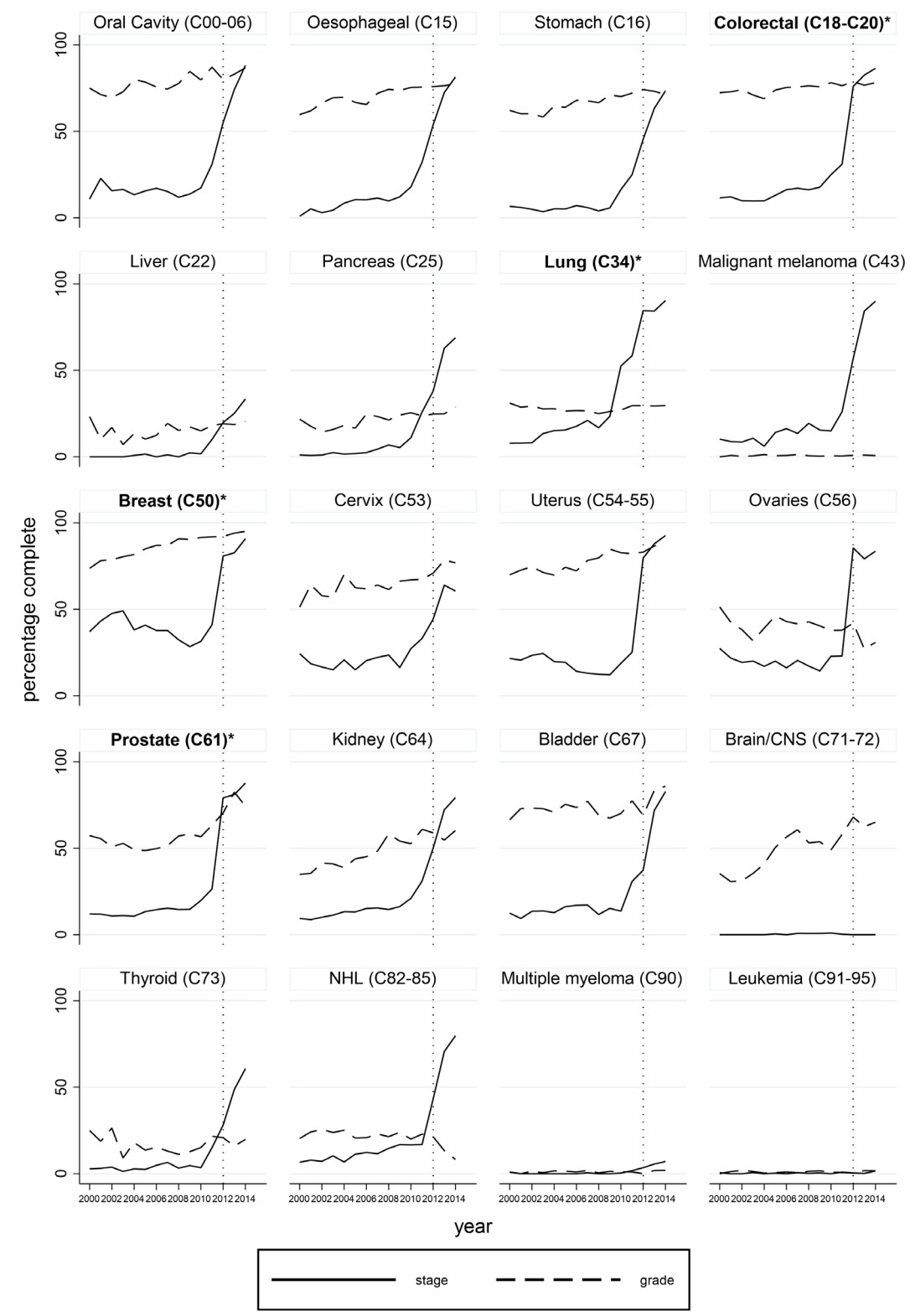

**Figure 5** Completeness of grade and stage for cancers identified using NCRAS data only. Cancer sites are ordered according to corresponding codes from the International Classification of Diseases, version 10 (ICD-10). *Four most common cancer sites. Grading information is not applicable to brain/CNS, sarcoma or haematological cancers and not required by in the national data standard (COSD) for prostate cancer. Core staging is not applicable to haematological and gynaecological cancers. Other types of staging are recommended by COSD. CNS, central nervous system; COSD, Cancer Outcomes and Services Data set; NCRAS, National Cancer Registration and Analysis Service; NHL, Non-Hodgkin's lymphoma.

only. Missing records may indicate that the patient did not receive that treatment modality or that the treatment modality was not recorded.

## DISCUSSION
### Statement of principal findings
We investigated the use of different sources of electronic health record data to identify incident cancers. For all case definitions, using individual or combined data sources, a minimum of 80% of incident site-specific cancers were confirmed using the gold standard algorithm; this rose to 94% of the four most common cancers. Use of cancer registration data alone or in any combination of data sources captured at least 80% of site-specific cancers identified by the gold standard algorithm, excepting bladder cancer, and 92% of cases for the four most common cancers. Combining CPRD GOLD, HES APC and ONS mortality data captured at least 80% of site-specific cancers excepting kidney, oral cavity and ovarian cancers, and captured >=89% of cases for the four most common cancers. Sensitivity was much more variable when using primary care or hospital data alone, and dropped to 65% when identifying bladder cancers using cancer registration data alone. Use of primary care or hospital data alone resulted in a small lag in identifying cancers of interest, compared with the gold standard dates but other case definitions captured cancers close to the gold standard date. Finally, while we observed minimal changes in PPVs and sensitivities between 2000 and 2014, completeness of NCRAS cancer registration stage and grade data increased markedly from 2012 onwards for specific cancer types, demonstrating that initiatives to improve data can have a profound impact on the quality of the data.[4] Completeness of cancer treatment recording was difficult to assess due to the absence of a missing category.

### Strengths and weaknesses of the study
The main strength of this study is that we have developed a gold standard algorithm using the entirety of the evidence available from CPRD to demonstrate the impact of choice of data sets in identifying incident cancers for real-life studies. We have also assessed the value of using NCRAS cancer registration data to measure stage, grade and cancer treatment modalities.

A limitation of the study is that our gold standard algorithm is not validated. We feel that we were justified in pre-weighting NCRAS data as more reliable that other data sources as NCRAS is a highly validated data set that matches, merges and quality checks data from multiple sources.[4] We did not consider NCRAS to be the outright gold standard as it is plausible that NCRAS does not identify all tumours diagnosed and treated in primary and secondary care. For most cancer sites, our gold standard algorithm identified a small proportion of cancers that are recorded in HES APC, CPRD GOLD or ONS mortality data but not in NCRAS. These

tumours may have been diagnosed and coded as invasive in primary or secondary care but not by NCRAS; been incorrectly coded in HES APC, CPRD GOLD or ONS mortality data; not have been notified to NCRAS (eg, tumours treated in private hospitals); or be the result of linkage errors between the data sets. The proportion of cancers identified in HES APC but not in NCRAS is particularly high for bladder cancer. This is likely to be the result of difficulties, inconsistencies and changes in the pathological definition and coding of cancers over time in NCRAS, which are greatest for bladder cancer.[4 14] This explanation is supported by the higher mortality rates that we observed in bladder cancer cases identified in NCRAS compared with other data sources. To identify incident cancers, we required 12 months of research quality follow-up in CPRD GOLD prior to inclusion in the study. Previous research has demonstrated that historic data is generally incorporated within the patient record with this time frame.[15] The identification of first ever cancers will also have been affected by different lengths of follow-up data available in linked data sources as NCRAS data collection started in 1990, HES APC in 1997 and ONS mortality data in 1998, and by the inclusion of all diagnostic codes in HES APC assuming that the first ever primary or secondary record identified incident cancer. Reassuringly, PPVs for liver and brain cancer were high for all individual and combinations of data sets suggesting that these were not unduly misclassified as primary incident cancers despite being common sites for metastases. Requiring internal confirmation within 6 months for cancers recorded in CPRD GOLD alone in our GOLD standard definition is more likely to discount cancers with poorer prognoses and those recorded in the last 6 months of follow-up. Our data cut only included NCRAS data for the top 20 cancers; earlier cancers at other sites will have been missed in this study.

It is also important to note that as the gold standard algorithm uses data recorded after the first record of the cancer site in any source (index date), it cannot be used to identify outcomes in applied studies and follow-up of cohort studies with cancer as an exposure would need to start at least 6 months after diagnosis; our first ever cancer record in any source definition would be more appropriate for most studies.

### Strengths and weaknesses in relation to other studies, discussing important differences in results
The most up-to-date study describing concordance between linked CPRD GOLD, HES APC and NCRAS data sets demonstrated that 2% to 4% of the five most common cancers recorded in CPRD GOLD are not confirmed in either HES APC or cancer registration data and 9% to 33% of registered cancers are not recorded in CPRD GOLD.[8] For cancers recorded in both sources, the diagnosis date was a median of 6 to 16 days later in CPRD GOLD than in the cancer registration data. Using CPRD GOLD alone to identify these

cancers marginally over-represented younger, healthier patients and identified 1% to 6% fewer deaths in the first 5 years after diagnosis. Use of HES APC only identified a higher proportion of patients with the correct diagnosis date than CPRD GOLD, but over-represented older patients and those diagnosed through the emergency route. The majority of registered cancers were picked up using both CPRD GOLD and HES APC (ranging from 91% for lung cancer to 97% for breast cancer). Previous research demonstrated similar results with substantial differences between cancer types.[5][6] Additionally, a study using data from 2001 to 2007 found that using HES data in addition to NCRAS data identified an additional 1.9%, 0.4% and 2.0% of surgically treated colorectal, lung and breast cancer cases, respectively.[16]

Our study is consistent with these results and provides more complete and practical evidence of the strengths and limitations of using individual and combinations of linked data sets to identify and characterise the 20 most common incident cancers.

We have also demonstrated the added value of using cancer registration data to measure stage and grade of incident cancers from about 2012 onwards. Levels of data completeness of staging information in the CPRD extract in 2012 were similar to those reported by the United Kingdom and Ireland Association of Cancer Registries (UKAICR).[9]

### Meaning of the study: possible explanations and implications for clinicians and policymakers

Use of NCRAS cancer registration data maximised the proportion of cases confirmed as true positive based on all available linked information and captured the highest proportion of true positive cases; highly complete staging and grading information is available from this source from approximately 2012. Case definitions based on a combination of CPRD GOLD, HES APC and ONS mortality data also had acceptable validity for the majority of cancer sites including the four most common cancers.

These findings should be considered when deciding which data sources to include in research studies and which sources to use to define cancer exposures, outcomes and covariates.

### UNANSWERED QUESTIONS AND FUTURE RESEARCH

Further research is required to investigate the validity of cancer recorded in CPRD GOLD and HES APC that are not recorded in the NCRAS data and to understand differences in cancer data recording with CPRD GOLD and CPRD Aurum, CPRD's recently launched primary care database based on records from practices that use EMIS software.[17] Further investigation would be required to confidently identify subtypes of cancer, either using codes available in each data set (eg, colon and rectal cancer) or additional information available in HES APC or NCRAS data. Use of NCRAS's recently launched Systemic Anti-Cancer Therapy (SACT)[18] and National Radiotherapy Data Sets will also improve ascertainment of therapies for future studies.

### CONCLUSION

Completeness and accuracy of recording of cancers in English data sources is high particularly when using NCRAS cancer registration data alone or in any combination with other data sources, and for the majority of cancers when using a combination of CPRD GOLD, HES APC and ONS mortality data. Completeness of cancer stage and grade variables in NCRAS was low before 2012 but appears to have substantially improved for most cancers in more recent calendar periods. It is not possible to validate completeness of the available treatment data; these should be used with caution. This study describes likely levels of misclassification for a range of data sources, combinations and cancer sites enabling cancer epidemiologists to optimise study design and better understand the limitations of their research.

**Author affiliations**
¹Department of Non-communicable Disease Epidemiology, London School of Hygiene & Tropical Medicine, London, UK
²Clinical Practice Research Datalink (CPRD), Medicines and Healthcare Products Regulatory Agency, London, UK

**Acknowledgements** This study is based in part on data from the Clinical Practice Research Datalink obtained under licence from the UK Medicines and Healthcare products Regulatory Agency. The data is provided by patients and collected by the National Health Service as part of their care and support. The interpretation and conclusions contained in this study are those of the author/s alone.

**Contributors** HS, RW and KB conceived the study and contributed to the study design. HS and KB did the data management. HS did the statistical analysis and wrote the first draft. HS, RW and KB contributed to subsequent drafts.

**Funding** CPRD funded access to the linked data sources used in this work. This work was additionally supported by the Wellcome Trust and Royal Society grant number 107731/Z/15/Z.

**Competing interests** RW is employed by Clinical Practice Research Datalink. HS and KB have academic honorary contracts at Public Health England for a separate collaborative research study.

**Patient consent for publication** Not required.

**Ethics approval** This study was approved by the London School of Hygiene & Tropical Medicine Ethics Committee (6202) and the Independent Scientific Advisory Committee for the Medicines and Healthcare products Regulatory Agency database research (12_068R).

**Provenance and peer review** Not commissioned; externally peer reviewed.

**Data availability statement** Data may be obtained from a third party and are not publicly available. Data were obtained from the Clinical Practice Research Datalink (CPRD), provided by the UK Medicines and Healthcare products Regulatory Agency. The authors' licence for using these data does not allow sharing of raw data with third parties. Information about access to CPRD data is available here: https://www.cprd.com/research-applications. Code lists for this study are available at https://doi.org/10.17037/data.00001519.

**ORCID iD**
Helen Strongman http://orcid.org/0000-0001-8739-4601

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
