## [Reviewer comments · BMJ Open]

ARTICLE DETAILS

TITLE (PROVISIONAL)	What are the implications of using individual and combined sources of routinely collected data to identify and characterise incident site-specific cancers? A concordance and validation study using linked English electronic health records data
AUTHORS	Strongman, Helen; Williams, Rachael; Bhaskaran, Krishnan

VERSION 1 – REVIEW

REVIEWER	David Goldsbury Cancer Council NSW, Australia
REVIEW RETURNED	14-Mar-2020

GENERAL COMMENTS	This is an interesting paper with a lot of useful and practical information for researchers who would like to identify cancer cases in England in their research, enabling, as noted in the conclusion, researchers to optimise study design and better understand the limitations of their research. The derived gold standard makes good use of the available information, however it might not be correct for all cancer types. There are several comments below but most are relatively minor and I think the paper is definitely worth pursuing. The main concern in establishing the gold standard is the relatively low sensitivity (65%) for bladder cancer using NCRAS data. Given the noted high level of validation of NCRAS data, should NCRAS alone be the gold standard? Is there an issue with case definition for bladder cancers in NCRAS compared with other sources, such as non-invasive/in situ lesions considered cancerous in the GP or hospital systems but not NCRAS? Or is the gold standard correct but the NCRAS missing or poorly categorising bladder cancers? The discussion of the missed bladder cancers should be expanded as this result casts doubt over the gold standard that is the basis for most of the results. The inclusion of treatment data, while interesting for other purposes, does not appear justified in this paper about the benefits and limitations of these data collections. There is no validation via comparison to a gold standard and there is no implicit level of completeness like with stage and grade information. This is briefly noted in the discussion section but leaving the results in there might be misleading without further justification. Similarly, the title referring to these being data to use in cancer epidemiology studies appears to over-state the results or at least not be specific enough to the study, which is about using these data to identify incident cancer cases.
--

	Can the authors add some discussion about how up-to-date and accessible the data collections are and whether there are any that are particularly good or difficult? Regarding the data coverage by time, please specify whether there are data for all collections for 1999-2014. It is mentioned in the discussion that there are different lengths of follow-up in the different sources, does this refer to data availability or is it a function of the case ascertainment methods? Also, the terms used to define the start of follow-up were unclear – what are the current registration date and up-to-standard date? Does the transfer out date mean someone has gone to another practice (perhaps identified with a record elsewhere) or left the area or something else? It is noted that cancers diagnosed at the start or end of the study period will not be as well enumerated. It is good to see a lead-in/wash-out period of 12 months is used, excluding anyone first recorded early on so that they are less likely to be those on extended treatment from much earlier or recurrences of earlier cancers. Can something similar be done at the end of follow-up for each person to create a complete-coverage analysis, at least as a sensitivity test? The description of “January 2017 CPRD build, set 13 linkage data” needs to be expanded or clarified. Are there regular builds/linkages of these datasets, using CPRD as a sampling frame? What is the level of coverage of the included GPs out of all in the UK? The inclusion of stage and grade information is interesting and a good example of how data quality improvements are possible. The authors should note where stage and/or grade is not relevant, such as stage for leukemia. There is no information relating to ethical approval for this study. In the abstract, it would be useful to include some numerical indication of the high completeness of staging data from 2012. Also, is there more information about the improvement in staging data? This is a dramatic improvement and would help other regions seeking to improve data quality. The introduction says using linked data “reduces sample size and has cost and logistical implications”, please give some detail/direction for these statements. Does it reduce sample size because it requires people to have records in multiple data collections? Clarify that the “completeness of cancer registry data” cited in reference 8 is the completeness of data fields within the available records, not the completeness of cases relative to all actual cases. What is an administrative cancer code? And further explanation is required for “no contradictions”. Should Figure 1 include ONS data in the contradictions? What is the CPRD derived death date and why is this used ahead of the ONS mortality data?
--	---

	There are a few references to “individual and different combinations of data”, or something similar. The word “different” could be removed here. The start of the results says that 6.8 million eligible people were “male and female”, this should be clarified. Can the observed differences in mortality according to method of ascertainment be generalised due to the nature of the relevant collections, driven by sensitivity and PPV? Will the CPRD GOLD-identified cases always have relatively lower mortality rates NCRAS-only cases have higher rates? If so, this should be noted as part of this useful practical application. The discussion note that a limitation is that the “analyses are limited to cancers diagnosed in England between 2000 and 2014”, but it is not clear why that is a limitation, even noting the changes in stage/grade data in 2012. Are EMIS practices different to the practices included in this study? In Figure 4 there are some unusual lines at the end of the plots, particularly for NHL. Why is this? Are they were very small numbers of people remain and single events cause large shifts? In Figure 5, is it expected that grade completeness is so low for liver, pancreas and lung cancers? Also, what is “cancerfull”?
--	---

REVIEWER	Una McMenamin Queen's University Belfast
REVIEW RETURNED	23-Mar-2020

GENERAL COMMENTS	The authors investigated the coverage in cancer diagnoses using individual and different combinations of linked English electronic health data, available for the English population, including the UK Clinical Practice Research Datalink primary care, National Cancer Registration and Analysis Service (NCRAS) cancer registration data, Hospital Episode Statistics (HES) and Office for National Statistics (ONS) death registration data. The authors implemented alternative case definitions to identify first site-specific cancers at the 20 most common cancer sites diagnosed between 2000-2014. Completeness and accuracy of recording of cancers was highest using NCRAS cancer registration data, either alone or in any combination with other data sources with the exception of bladder cancer. Completeness was lower but largely similar when using a combination of CPRD GOLD, HES APC and ONS mortality data, especially with respect to the most common cancers. Completeness however was lower for kidney, oral cavity and ovarian cancer. The authors also investigated the completeness of cancer stage and grade within NCRAS and found that this was low before 2012 but improved for most cancers in more recent years. The study provides some quantification as to the likely levels of misclassification of incidence cancers for a range of commonly used data sources and includes a comprehensive list of cancer sites. The findings therefore will allow cancer epidemiologists to make more informed decisions as to which datasets (or
--

	combination) would be most appropriate for their particular investigation. The authors are to be commended for their comprehensive analysis including a number of sub-group analyses according to sex, age and calendar year. I have however listed a number of points and suggestions below for review. 1. In the introduction (and discussion), the authors should further clarify the differences between their study and the recent study conducted by Arhi et al. (2018), especially as similar time frames were used. Further emphasise is required as to the additional value that this study brings. The authors should also include the Ahri et al. (2018) study in their Introduction when discussing previous evidence.2. Did the authors make any efforts to minimise the potential for misclassification of common metastatic cancer sites that could be misclassified as incidence cancers, e.g. liver, brain?3. Is there any evidence that the authors could cite to demonstrate that the combination of these types of datasets is the gold-standard, e.g. comparison to clinical audits?4. Can the authors comment on their justification for identifying incident cancers recorded in any diagnosis field for HES APC as opposed to identifying cancers based on the primary hospital diagnosis? Is there any evidence to demonstrate the reliability of this approach?5. The authors required all eligible patients to have at least 1 year of CPRD records to be included in the analysis, did the authors consider conducting a sensitivity analysis extending this to time period (e.g. 5 years) which would also provide more confidence that patients did not have a prior history of cancer.6. Can the authors comment on the potential reasons as to why there would be cancers not present in NCRAS, but present only in HES or CPRD? This would be useful considering that cancer registration is generally deemed to be the gold-standard in the identification of incident cancers.7. Can the authors comment on the applicability of these results for sub-types of cancer - certain cancers are now recognised as two distinct cancer types in terms of aetiology and treatment (e.g. oesophageal adenocarcinoma and squamous cell carcinoma, or colon and rectal cancer). There is likely insufficient information in HES and CPRD to allow for stratification based on anatomical location or morphology but this warrants discussion given that cancer epidemiologists require this level of detail to conduct meaningful studies into these particular cancer types.8. If possible, it would be useful if the abstract contained a summary of the discrepancies in the cancer diagnoses dates as this is an essential consideration in cancer epidemiology studies.9. Throughout the manuscript, the authors could be more specific about which set of combinations of datasets they are referring to as it was difficult to follow at times.
--	---

	10. The conclusions should include the caveat that the ascertainment of incident cases using a combination of primary care, hospitalisation and death registration data was suboptimal for some cancer sites. 11. The abstract and methods should specify more clearly that this analysis is restricted to datasets within England and are not applicable to the UK as a whole. 12. Line 170 – what does ‘set 13’ refer to?
--	--

VERSION 1 – AUTHOR RESPONSE

Reviewer: 1

Reviewer Name: David Goldsbury

Institution and Country: Cancer Council NSW, Australia

Please state any competing interests or state ‘None declared’: None declared

Please leave your comments for the authors below

This is an interesting paper with a lot of useful and practical information for researchers who would like to identify cancer cases in England in their research, enabling, as noted in the conclusion, researchers to optimise study design and better understand the limitations of their research. The derived gold standard makes good use of the available information, however it might not be correct for all cancer types. There are several comments below but most are relatively minor and I think the paper is definitely worth pursuing.

Author response: Many thanks for your positive review of our manuscript.

Comment 1

The main concern in establishing the gold standard is the relatively low sensitivity (65%) for bladder cancer using NCRAS data. Given the noted high level of validation of NCRAS data, should NCRAS alone be the gold standard? Is there an issue with case definition for bladder cancers in NCRAS compared with other sources, such as non-invasive/in situ lesions considered cancerous in the GP or hospital systems but not NCRAS? Or is the gold standard correct but the NCRAS missing or poorly categorising bladder cancers? The discussion of the missed bladder cancers should be expanded as this result casts doubt over the gold standard that is the basis for most of the results.

Author response: We have looked into the reasons why cancers may be recorded in HES APC and CPRD GOLD data but not NCRAS. We couldn’t find any published audit reports or validation studies but did find some clues in the HES APC resource profile. This noted that for bladder cancer in particular, there have been inconsistencies and changes in the pathological definition and coding of invasive cancers over time in NCRAS. Tumours may therefore have been diagnosed and classified as invasive in HES APC, CPRD GOLD or ONS mortality but not in NCRAS. Other reasons for missing cancer records in NCRAS are that cancers may have been incorrectly coded in HES APC or NCRAS, not been notified to NCRAS (e.g. treated and paid for privately), or be the result of linkage errors between datasets. We have reworded the limitations section of the discussion and last bullet of the strengths and limitations section to reflect this.

Author change:

Added: (p3, line 59) A key limitation was that our gold standard algorithm is not validated and may be affected by differences in clinical diagnosis and coding of invasive cancers between data sources.

Before: (p11, line 250) Another limitation is that our gold standard algorithm pre-weighted NCRAS data as more reliable than other data sources. We feel this is justified as NCRAS is a highly validated data set that matches and merges data from multiple sources⁴. However, this decision will have given case definitions involving NCRAS an inherent advantage in measures of positive predictive value and sensitivity.

After: (p11, line 261) A limitation of the study is that our gold standard algorithm is not validated. We feel that we were justified in pre-weighting NCRAS data as more reliable than other data sources as NCRAS is a highly validated data set that matches, merges and quality checks data from multiple sources⁴. We did not consider NCRAS to be the outright gold standard as it is plausible that NCRAS does not identify all tumours diagnosed and treated in primary and secondary care. For most cancer sites, our gold standard algorithm identified a small proportion of cancers that are recorded in HES APC, CPRD GOLD or ONS mortality data but not in NCRAS. These tumours may have been diagnosed and coded as invasive in primary or secondary care but not by NCRAS; been incorrectly coded in HES APC, CPRD GOLD or ONS mortality data; not have been notified to NCRAS (e.g. tumours treated in private hospitals); or be the result of linkage errors between the data sets. The proportion of cancers identified in HES APC but not in NCRAS is particularly high for bladder cancer. This is likely to be the result of difficulties, inconsistencies and changes in the pathological definition and coding of cancers over time in NCRAS, which are greatest for bladder cancer^{4,14}. This explanation is supported by the higher mortality rates that we observed in bladder cancer cases identified in NCRAS compared with other data sources.

Comment 2

The inclusion of treatment data, while interesting for other purposes, does not appear justified in this paper about the benefits and limitations of these data collections. There is no validation via comparison to a gold standard and there is no implicit level of completeness like with stage and grade information. This is briefly noted in the discussion section but leaving the results in there might be misleading without further justification.

Author response: We agree that the treatment data cannot be validated but think that it is important that researchers understand that these data are unlikely to be complete. We've added a sentence to the conclusion to re-iterate this.

Author change: (p15, line 340) "It is not possible to validate completeness of the available treatment data; these should be used with caution."

Comment 3

Similarly, the title referring to these being data to use in cancer epidemiology studies appears to overstate the results or at least not be specific enough to the study, which is about using these data to identify incident cancer cases.

Author response: We have adapted the title in response to your comment and those from the editors and reviewer 2.

Author change:

Before: Benefits and limitations of using individual and different combinations of linked English routine data sources in cancer epidemiology studies

After: What are the implications of using individual and combined sources of routinely collected data to identify and characterise incident site-specific cancers? A concordance and validation study using linked English electronic health records data.

Comment 4

Can the authors add some discussion about how up-to-date and accessible the data collections are and whether there are any that are particularly good or difficult?

Author response: We agree that this is an important consideration when deciding which combination of datasets to use and have briefly commented on it in the introduction including an independent citation to provide more detail. We missed the point about coverage periods and have now added add this to the introduction (see below) and methodology (see comment 5).

Author change:

Before: (p3, line 68) Use of linked data reduces sample size and has cost and logistical implications, which are greatest for NCRAS data. Research teams therefore commonly choose not to use all available linked data³.

After: (p3, line 69) Use of linked data reduces the sample to the common source population and data coverage period for each included dataset and has cost and logistical implications, which are greatest for NCRAS data. Research teams therefore commonly choose not to use all available linked data³. [see also comment 11]

Comment 5

Regarding the data coverage by time, please specify whether there are data for all collections for 1999-2014. It is mentioned in the discussion that there are different lengths of follow-up in the different sources, does this refer to data availability or is it a function of the case ascertainment methods? Also, the terms used to define the start of follow-up were unclear – what are the current registration date and up-to-standard date? Does the transfer out date mean someone has gone to another practice (perhaps identified with a record elsewhere) or left the area or something else?

Author response: CPRD release monthly builds of the primary care data including all data submitted by general practices prior to processing of the build. Follow-up for individual patients is determined by registration with the practice delimited by the current registration date and transfer out date and by the CPRD estimated start and known end of continuous data collection by the practice (up-to-standard date and last collection date). The most common reason for changing a practice is moving to a different area.

Linkage datasets are released on a less regular basis and include all patients whose identifiers were sent to the trusted third party prior to linkage processing. The coverage period of the linked datasets often limits the study period for studies. The latest start of data coverage period for datasets used in the study was 02/02/1998 for ONS mortality data and the earliest end of data coverage was 31/12/2014 for ONS mortality data. We have clarified this below including reinserting reference 2 which describes the linkage process.

Author change:

Before: (Page 4, row 112) Start of follow-up was defined as the latest of the current registration date within the practice and the practice up-to-standard date, and end of follow-up as the earliest of the patient transfer out date, CPRD derived death date, or practice last collection date.

After: Start of follow-up was defined as the latest of the current registration date within the practice and the CPRD estimated start of continuous data collection for the practice (up-to-standard date). End of follow-up was determined by the earliest of the date the patient left the practice, ONS mortality date of death, or practice last collection date.

Before: (p5, line 95). We completed a concordance study using linked CPRD GOLD, HES APC, ONS mortality and NCRAS data (January 2017 CPRD build, set 13 linkage data, study period 1 Jan 2000 – 31 December 2014).

After: (p5, line 98). We completed a concordance study using linked 12 English CPRD GOLD, HES APC, ONS mortality and NCRAS data. CPRD GOLD data were extracted from the January 2017 monthly release and the 13th update to CPRD's linked data. The study period was 1 Jan 2000 – 31 December 2014, with 31 December matching the end of the NCRAS coverage period.

Before: (p11, line 254) The algorithm will also have been affected by different lengths of follow-up data available in the different data sources. For example, NCRAS data collection started later than CPRD GOLD and HES which may account for some of the misclassification of incident cases when using NCRAS alone.

After: (p11, line 277) The identification of first ever cancers will also have been affected by different lengths of follow-up data available in linked data sources as NCRAS data collection started in 1990, HES APC in 1997 and ONS mortality data in 1998,

Comment 6

It is noted that cancers diagnosed at the start or end of the study period will not be as well enumerated. It is good to see a lead-in/wash-out period of 12 months is used, excluding anyone first recorded early on so that they are less likely to be those on extended treatment from much earlier or recurrences of earlier cancers. Can something similar be done at the end of follow-up for each person to create a complete-coverage analysis, at least as a sensitivity test?

Author response: We used a 12 month lead in period to make sure that we recorded incident rather than prevalent cancers. We are confident that cancers towards the end of follow-up will be captured.

Author change: No change

Comment 7

The description of "January 2017 CPRD build, set 13 linkage data" needs to be expanded or clarified. Are there regular builds/linkages of these datasets, using CPRD as a sampling frame? What is the level of coverage of the included GPs out of all in the UK?

Author response: See comment 5

Comment 8

The inclusion of stage and grade information is interesting and a good example of how data quality improvements are possible. The authors should note where stage and/or grade is not relevant, such as stage for leukemia.

Author response: We have added information about cancer registration requirements for grading and staging data to the figure footnote. These are based on the COSD dataset which we've now mentioned in the methodology.

Author change:

Before: (p5, line 105) National cancer registration data are collected by NCRAS which is part of Public Health England (PHE).

After: (p5, line 110) National cancer registration data are collected by NCRAS which is part of Public Health England (PHE) in accordance with the Cancer Outcomes and Services Dataset (COSD)¹³ which has been the national standard for reporting of cancer in England since January 2013.

Before: (Figure 5 legend) Cancer sites are ordered according to corresponding codes from the International Classification of Diseases, version 10 (ICD-10). *Four most common cancer sites. NHL = Non hodgkin lymphoma.

After: (Figure 5 legend) Cancer sites are ordered according to corresponding codes from the International Classification of Diseases, version 10 (ICD-10). *Four most common cancer sites. NHL = Non hodgkin lymphoma. Grading information is not applicable to brain/CNS, sarcoma or haematological cancers and not required by in the national data standard (COSD) for prostate cancer. Core staging is not applicable to haematological and gynaecological cancers. Other types of staging are recommended by COSD.

Comment 9

There is no information relating to ethical approval for this study.

Author response: We have added information about ethical approval for this study in the methodology.

Author change:

Added (p7, line 177): This study was approved by the London School of Hygiene & Tropical Medicine Ethics Committee (6202) and the Independent Scientific Advisory Committee for the Medicines and Healthcare products Regulatory Agency database research (12_068R).

Comment 10

In the abstract, it would be useful to include some numerical indication of the high completeness of staging data from 2012. Also, is there more information about the improvement in staging data? This is a dramatic improvement and would help other regions seeking to improve data quality.

Author response: We have added some numerical indication of the high completeness of staging data from 2012 to the abstract. This improvement was a result of the migration of all regional cancer registries into a national service, and the accompanying changes to processes, software etc. There was also a focused effort by PHE's data liaison teams who work with NHS trusts to improve their submitted data. More information is available in reference 4 (NCRAS cancer registration data resource profile).

Author change:

Before: (p2, line 42) Completeness of staging data in cancer registration data was high from 2012

After: (p2, line 42) Completeness of staging data in cancer registration data was high from 2012
Completeness of staging data in cancer registration data was high from 2012 (minimum 76.0% 2012
86.4% 2014 for the four most common cancers).

Before: (p11, line 248) However, substantial improvements in completeness of stage and grade data in 2012 demonstrate that initiatives to improve data can have a profound impact on the quality of data.

After: (p11, line 251) completeness of NCRAS cancer registration stage and grade data increased markedly from 2012 onwards for specific cancer types, demonstrating that initiatives to improve data can have a profound impact on the quality of the data⁴.

Comment 11

The introduction says using linked data “reduces sample size and has cost and logistical implications”, please give some detail/direction for these statements. Does it reduce sample size because it requires people to have records in multiple data collections?

Author response: Using linked data reduces sample size because the sample has to be restricted to the source population and coverage period for each linked dataset. This is discussed on reference 3. The first line of our results section (p8, line 169) quantifies the impact of reducing the CPRD GOLD dataset to patients who were eligible for linkage in this study.

Author change:

Before: (p3, line 68) Use of linked data reduces sample size and has cost and logistical implications, which are greatest for NCRAS data. Research teams therefore commonly choose not to use all available linked data³.

After: (p3, line 69) Use of linked data reduces the sample to the common source population and data coverage period for each included dataset and has cost and logistical implications, which are greatest for NCRAS data. Research teams therefore commonly choose not to use all available linked data³.

Comment 12

Clarify that the “completeness of cancer registry data” cited in reference 8 is the completeness of data fields within the available records, not the completeness of cases relative to all actual cases.

Author change:

Before: (p4, line 80) National data are available describing completeness of cancer registry data in each collection year⁸

After: (p4, line 82) National data are available describing completeness of data fields within the cancer registry data in each collection year⁸

Comment 13

What is an administrative cancer code? And further explanation is required for “no contradictions”.

Author response: Administrative cancer codes are entered into GP data to record care events such as cancer care reviews, administration of chemotherapy and radiotherapy, and referrals for suspected

cancers. The majority of confirmatory codes identified for this study were for cancer care reviews. We have included more information about this in the legend for figure 1. We have also provided a brief description of a contradiction early in our description of the algorithm. More detailed information is available in figure 1.

Author change:

Before: (Figure 1 legend) * Includes administrative codes and historical cancers.

After: (Figure 1 legend) * Includes administrative codes (e.g. cancer care review, chemotherapy / radiotherapy, referrals for suspected cancer) and historical cancers.

Before: (p6, line 138) We developed a gold standard algorithm that classifies incident records of the 20 most common cancers by comparing the first malignant cancer identified in each individual source (Figure 1). Cancers recorded in NCRAS alone with no contradictions were considered true cases whereas cancers recorded in HES APC alone or GOLD alone required internal confirmation within that source in the form of another code for cancer consistent with the same site (or with site unspecified) within 6 months and no contradictory codes (e.g. for cancers at other sites) in this period. Where cancer records were present in >1 data source, we considered a site-specific cancer to be a true case (a) if it was recorded as the first cancer in NCRAS and the total number of data sources with records for cancer at that site was equal to or greater than the number of data sources with contradictory records (i.e. records for first cancers at different sites); or (b) where the cancer was not present in NCRAS, if there were more data sources in total with records for cancer at that site than data sources with contradictory records.

After: (p7, line 146) We developed a gold standard algorithm that classifies incident records of the 20 most common cancers by comparing the first malignant cancer identified in each individual source (Figure 1). Cancers recorded in NCRAS alone with no contradictions (i.e. records for first cancers at different sites) were considered true cases whereas cancers recorded in HES APC alone or GOLD alone required internal confirmation within that source in the form of another code for cancer consistent with the same site (or with site unspecified) within 6 months and no contradictory codes (e.g. for cancers at other sites) in this period. Where cancer records were present in >1 data source, we considered a site-specific cancer to be a true case (a) if it was recorded as the first cancer in NCRAS and the total number of data sources with records for cancer at that site was equal to or greater than the number of data sources with contradictory records (i.e. records for first cancers at different sites); or (b) where the cancer was not present in NCRAS, if there were more data sources in total with records for cancer at that site than data sources with contradictory records.

Comment 14

Should Figure 1 include ONS data in the contradictions?

Author response: We did not include ONS data in the contradictions because we would not expect the death date to be similar to the diagnosis date in other sources.

Comment 15

What is the CPRD derived death date and why is this used ahead of the ONS mortality data?

Author response: The CPRD derived death date uses data recorded in several places in the GP record. On reflection we decided to replace this with the ONS mortality date throughout the analysis. As expected, this had minimal impact on the sample sizes and results.

Author change:

Before: (p5, line 114) end of follow-up as the earliest of the patient transfer out date, CPRD derived death date, or practice last collection date.

After: (p5, line 121) End of follow-up was determined as the data the patient left the practice, ONS mortality date of death, or practice last collection date.

Before: (p7, line 161) We used Kaplan-Meier methods to describe mortality over time for cancers identified using each definition. The CPRD derived death date was used for these analyses.

After: (p8, line 170) We used Kaplan-Meier methods to describe mortality over time for cancers identified using each definition. The ONS mortality death date was used for these analyses.

Comment 16

There are a few references to “individual and different combinations of data”, or something similar. The word “different” could be removed here.

Author response: We have removed “different” from the title, lines 20, 53 and 228.

Comment 17

The start of the results says that 6.8 million eligible people were “male and female”, this should be clarified.

Author response: A minimal proportion of individuals recorded in CPRD GOLD have gender recorded as indeterminate. It is difficult to differentiate between true cases of indeterminate sex and unknowns. These people are therefore generally removed from research studies.

Author change:

Before: (p8, line 169) Of 14 747 047 research quality patients in the CPRD GOLD January 2017 build, 8 893 326 were eligible for linkage to HES, ONS mortality and NCRAS data in set 13; 6 791 074 of these were male and female and had at least one year of follow-up between 1 January 1999 and 31 December 2014 and were included in the study population.

After: (p8, line 181) Of 14 747 047 research quality patients in the CPRD GOLD January 2017 build, 8 893 326 were eligible for linkage to HES, ONS mortality and NCRAS data in set 13; 237 were excluded due to unknown sex. Of the remainder, 6 791 074 had at least one year of follow-up between 1 January 1999 and 31 December 2014 and were included in the study population.

Comment 18

Can the observed differences in mortality according to method of ascertainment be generalised due to the nature of the relevant collections, driven by sensitivity and PPV? Will the CPRD GOLD-identified cases always have relatively lower mortality rates NCRAS-only cases have higher rates? If so, this should be noted as part of this useful practical application.

Author response: For the majority of cancers, we observed minimal differences in mortality according to method of ascertainment. The slightly lower mortality rates for cases ascertained in CPRD GOLD only compared to NCRAS only are likely to be due to misclassification of suspected and benign cancers as malignant cancers in CPRD GOLD and to a lesser extent, earlier stage malignant

cancers not being recorded in NCRAS data. This explanation is supported by the more pronounced differences in mortality observed for bladder cancer which is linked to known differences in the diagnosis and recording of invasive cancers between datasets.

Author change:

Added: (p12, line 270) The proportion of cancers identified in HES APC but not in NCRAS is particularly high for bladder cancer. This is likely to be the result of difficulties, inconsistencies and changes in the pathological definition and coding of cancers over time in NCRAS, which are greatest for bladder cancer^{4,14}. This explanation is supported by the higher mortality rates that we observed in bladder cancer cases identified in NCRAS compared with other data sources.

Comment 19

The discussion note that a limitation is that the “analyses are limited to cancers diagnosed in England between 2000 and 2014”, but it is not clear why that is a limitation, even noting the changes in stage/grade data in 2012.

Author response: We have removed this as a limitation and moved linked information to earlier in the discussion.

Author change:

Deleted: (p11, line 246) A limitation of the study is that our analyses are limited to cancers diagnosed in England between 2000 and 2014. We observed minimal changes in PPVs and sensitivities over this time period suggesting that our findings are generalisable to later years. However, substantial improvements in completeness of stage and grade data in 2012 demonstrate that initiatives to improve data can have a profound impact on the quality of data.

Before: (p10, line 237) Finally, we found that completeness of NCRAS cancer registration stage and grade data increased markedly from 2012 onwards and for specific cancer types; completeness of cancer treatment recording was difficult to assess due to the absence of a missing category.

After: (p11, line 250) Finally, whilst we observed minimal changes in PPVs and sensitivities between 2000 and 2014, completeness of NCRAS cancer registration stage and grade data increased markedly from 2012 onwards for specific cancer types, demonstrating that initiatives to improve data can have a profound impact on the quality of the data⁴. Completeness of cancer treatment recording was difficult to assess due to the absence of a missing category.

Comment 20

Are EMIS practices different to the practices included in this study?

Author response: The CPRD GOLD and CPRD Aurum databases use data collected from practices that use Vision end EMIS software respectively. These software systems are both designed to support clinical care in NHS general practices and follow inter-operability requirements set out by NHS Digital. However, differences in how general practices enter data into the two systems may affect data comparability. This is discussed in the referenced data resource profile.

Author change:

Before: (p13, line 299) Further research is required to understand differences in cancer data recording with CPRD GOLD and CPRD Aurum, CPRD's recently launched primary care database based on records from EMIS practices¹⁴.

After: (p14, line 327) Further research is required to ... and to understand differences in cancer data recording with CPRD GOLD and CPRD Aurum, CPRD's recently launched primary care database based on records from practices that use EMIS software¹⁷.

Comment 21

In Figure 4 there are some unusual lines at the end of the plots, particularly for NHL. Why is this? Are they were very small numbers of people remain and single events cause large shifts?

Author response: These unusual lines appear to have been driven by noise in the CPRD death date information that we were originally using. This has been fixed after we re-ran the analysis using ONS death date in place of CPRD death date, in response to other reviewer comments.

Comment 22

In Figure 5, is it expected that grade completeness is so low for liver, pancreas and lung cancers? Also, what is "cancerfull"?

Author response: The best comparative data we can find describes completeness of grading data for individual tumours over the period 1995-2016 https://www.cancerdata.nhs.uk/explorer/tumour_types . Completeness was 16.8%, 20.2% and 27.3% for liver, pancreas and lung cancer respectively which is consistent with our graphs.

Cancerfull is the variable name in our dataset that was automatically added to the graph by stata. We did not intend to show this and have removed it from the graph

Reviewer: 2

Reviewer Name: Úna McMEnamin

Institution and Country: Queen's University Belfast

Please state any competing interests or state 'None declared': None declared.

Please leave your comments for the authors below

The authors investigated the coverage in cancer diagnoses using individual and different combinations of linked English electronic health data, available for the English population, including the UK Clinical Practice Research Datalink primary care, National Cancer Registration and Analysis Service (NCRAS) cancer registration data, Hospital Episode Statistics (HES) and Office for National Statistics (ONS) death registration data. The authors implemented alternative case definitions to identify first site-specific cancers at the 20 most common cancer sites diagnosed between 2000-2014.

Completeness and accuracy of recording of cancers was highest using NCRAS cancer registration data, either alone or in any combination with other data sources with the exception of bladder cancer. Completeness was lower but largely similar when using a combination of CPRD GOLD, HES APC and ONS mortality data, especially with respect to the most common cancers. Completeness however was lower for kidney, oral cavity and ovarian cancer. The authors also investigated the completeness of cancer stage and grade within NCRAS and found that this was low before 2012 but improved for most cancers in more recent years.

The study provides some quantification as to the likely levels of misclassification of incidence cancers for a range of commonly used data sources and includes a comprehensive list of cancer sites. The

findings therefore will allow cancer epidemiologists to make more informed decisions as to which datasets (or combination) would be most appropriate for their particular investigation. The authors are to be commended for their comprehensive analysis including a number of sub-group analyses according to sex, age and calendar year.

Author response: Thank you for your positive review of our manuscript.

I have however listed a number of points and suggestions below for review.

1. In the introduction (and discussion), the authors should further clarify the differences between their study and the recent study conducted by Arhi et al. (2018), especially as similar time frames were used. Further emphasise is required as to the additional value that this study brings. The authors should also include the Ahri et al. (2018) study in their Introduction when discussing previous evidence.

Author response: We have cited Arhi in the introduction and clarified that this study was restricted to the five most common cancers and that all previous studies focused on concordance between CPRD GOLD and NCRAS. This new study therefore provides a more complete assessment of the benefits and limitations of using different combinations of data sources within the context of practical designs.

Author change:

Before: (p4, line 74) Validation studies assessing concordance between CPRD GOLD, HES APC and NCRAS data have estimated high Positive Predictive Values (PPVs) for CPRD GOLD data and varying proportions of registered cancers that are not captured in CPRD GOLD and HES APC⁵⁻⁷. These studies have focused on the most common cancers and concordance between CPRD GOLD only and NCRAS, and do not provide a complete assessment of the benefits and limitations of using different combinations of data sources.

After: (p4, line 76) Validation studies assessing concordance between CPRD GOLD, HES APC and NCRAS data have estimated high Positive Predictive Values (PPVs) for CPRD GOLD data and varying proportions of registered cancers that are not captured in CPRD GOLD and HES APC⁵⁻⁸. The most up to date analysis by Arhi et al. included the 5 most common cancers and all papers focused on concordance between CPRD GOLD only and NCRAS; existing evidence therefore does not provide a complete assessment of the benefits and limitations of using different combinations of data sources within the context of practical study designs.

Before: (p12, line 280) Our study is consistent with these results and provides more complete evidence for a wide range of cancers which will allow researchers to understand the strengths and limitations of different study designs.

After: (p13, line 309) Our study is consistent with these results and provides more complete and practical evidence of the strengths and limitations of using individual and combinations of linked datasets to identify and characterise the twenty most common incident cancers.

2. Did the authors make any efforts to minimise the potential for misclassification of common metastatic cancer sites that could be misclassified as incidence cancers, e.g. liver, brain?

Author response: We took measures to limit our study definitions to site-specific primary incident cancers, including requiring a wash out period of 12 months before patients were included in the study and limiting our code lists to codes for primary cancers. Reassuringly PPVs for liver and brain cancer were high for all individual and combinations of datasets suggesting that these were not

unduly misclassified despite being common sites for metastases. We have also run ad-hoc analyses following your query and found that a minimal proportion of individuals with liver and brain cancers identified in CPRD GOLD and / or HES APC but not in NCRAS data had an alternative cancer recorded in NCRAS. We have added a sentence about this to the discussion (see also change in response to comment 5 which groups responses to all comments related to the identification of incident cancers).

Author change:

Added: (12, line 281) Reassuringly, PPVs for liver and brain cancer were high for all individual and combinations of datasets suggesting that these were not unduly misclassified as primary incident cancers despite being common sites for metastases.

3. Is there any evidence that the authors could cite to demonstrate that the combination of these types of datasets is the gold-standard, e.g. comparison to clinical audits?

Author response: We have modified the discussion to justify our decision making within the context of options that are widely available to electronic health researchers using English data, and to describe potential limitations. We have also changed the last bullet of the strengths and limitations section to reflect this.

Author change:

Added: (p3, line 59) A key limitation was that our gold standard algorithm is not validated and may be affected by differences in clinical diagnosis and coding of invasive cancers between data sources.

Before: (p11, line 250) Another limitation is that our gold standard algorithm pre-weighted NCRAS data as more reliable than other data sources. We feel this is justified as NCRAS is a highly validated data set that matches and merges data from multiple sources⁴. However, this decision will have given case definitions involving NCRAS an inherent advantage in measures of positive predictive value and sensitivity.

After: (p11, line 261) A limitation of the study is that our gold standard algorithm is not validated. We feel that we were justified in pre-weighting NCRAS data as more reliable than other data sources as NCRAS is a highly validated data set that matches, merges and quality checks data from multiple sources⁴. We did not consider NCRAS to be the outright gold standard as it is plausible that NCRAS does not identify all tumours diagnosed and treated in primary and secondary care. For most cancer sites, our gold standard algorithm identified a small proportion of cancers that are recorded in HES APC, CPRD GOLD or ONS mortality data but not in NCRAS. These tumours may have been diagnosed and coded as invasive in primary or secondary care but not by NCRAS; been incorrectly coded in HES APC, CPRD GOLD or ONS mortality data; not have been notified to NCRAS (e.g. tumours treated in private hospitals); or be the result of linkage errors between the data sets. The proportion of cancers identified in HES APC but not in NCRAS is particularly high for bladder cancer. This is likely to be the result of difficulties, inconsistencies and changes in the pathological definition and coding of cancers over time in NCRAS, which are greatest for bladder cancer^{4,14}. This explanation is supported by the higher mortality rates that we observed in bladder cancer cases identified in NCRAS compared with other data sources.

4. Can the authors comment on their justification for identifying incident cancers recorded in any diagnosis field for HES APC as opposed to identifying cancers based on the primary hospital diagnosis? Is there any evidence to demonstrate the reliability of this approach?

Author response: There is no clear evidence or guidelines demonstrating the reliability of using either primary diagnostic codes only or all codes to identify incident cancers in HES. The primary hospital diagnosis field indicates the primary cause of admission for each consultant episode whereas the secondary fields include pre-existing conditions that affect the patient's care in hospital and conditions that were diagnosed while the patient was in hospital. We have assumed that both primary and secondary diagnostic codes relate to incident cancers if there is no previous record of cancer.

Author change: see response to comment 5 which groups responses to all comments related to the identification of incident cancers.

5. The authors required all eligible patients to have at least 1 year of CPRD records to be included in the analysis, did the authors consider conducting a sensitivity analysis extending this to time period (e.g. 5 years) which would also provide more confidence that patients did not have a prior history of cancer.

Author response: Previous research has demonstrated most catch-up recording of medical history happens within the first year. We have highlighted this issue in the discussion and cited the relevant paper.

Author change:

Before: (p11, line 254) The algorithm will also have been affected by different lengths of follow-up data available in the different data sources. For example, NCRAS data collection started later than CPRD GOLD and HES which may account for some of the misclassification of incident cases when using NCRAS alone.

After: (p12, line 275) To identify incident cancers, we required 12 months of research quality follow-up in CPRD GOLD prior to inclusion in the study. Previous research has demonstrated that historic data is generally incorporated within the patient record with this time frame¹⁵ The identification of first ever cancers will also have been affected by different lengths of follow-up data available in linked data sources as NCRAS data collection started in 1990, HES APC in 1997 and ONS mortality data in 1998, and by the inclusion of all diagnostic codes in HES APC assuming that the first ever primary or secondary record identified incident cancer. Reassuringly, PPVs for liver and brain cancer were high for all individual and combinations of datasets suggesting that these were not unduly misclassified as primary incident cancers despite being common sites for metastases.

6. Can the authors comment on the potential reasons as to why there would be cancers not present in NCRAS, but present only in HES or CPRD? This would be useful considering that cancer registration is generally deemed to be the gold-standard in the identification of incident cancers.

Author response: We have investigated potential reasons as to why there would be cancers not present in NCRAS, but present only in HES or CPRD. Reasons include that tumours may have been diagnosed and coded as invasive in primary or secondary care but not by NCRAS; been incorrectly coded in HES APC, CPRD GOLD or ONS mortality data; not have been notified to NCRAS (e.g. tumours treated in private hospitals); or be the result of linkage errors between the data sets. We have added this information to the discussion (see response to comment 3). We have also mentioned the need for further research to investigate the validity of cancers recorded in CPRD GOLD and HES APC that are not recorded in NCRAS data (see below).

Author change:

Before: (p13, line 299) Further research is required to understand differences in cancer data recording with CPRD GOLD and CPRD Aurum, CPRD's recently launched primary care database based on records from EMIS practices¹⁴.

After: (p14, line 327) Further research is required to investigate the validity of cancer recorded in CPRD GOLD and HES APC that are not recorded in the NCRAS data and to understand differences in cancer data recording with CPRD GOLD and CPRD Aurum, CPRD's recently launched primary care database based on records from practices that use EMIS software¹⁷.

7. Can the authors comment on the applicability of these results for sub-types of cancer - certain cancers are now recognised as two distinct cancer types in terms of aetiology and treatment (e.g. oesophageal adenocarcinoma and squamous cell carcinoma, or colon and rectal cancer). There is likely insufficient information in HES and CPRD to allow for stratification based on anatomical location or morphology but this warrants discussion given that cancer epidemiologists require this level of detail to conduct meaningful studies into these particular cancer types.

Author response: It is generally possible to distinguish between subtypes of in all datasets where these are represented by different 3-digit ICD-10 codes (e.g. C18 colon cancer, C20 rectal cancer). Care must be taken where unspecified codes exist, as these may be more common in CPRD GOLD and HES APC than in the NCRAS data. An example is C85 (other and unspecified types of NHL). Further subdivisions could be identified using the 4-digit ICD-10 codes available with HES APC or additional fields in the NCRAS data. We've added a comment on this to the unanswered questions and future research section of the discussion.

Author change:

Added: (p14, line 330) Further investigation would be required to confidently identify subtypes of cancer, either using codes available in each dataset (e.g. colon and rectal cancer) or additional information available in HES APC or NCRAS data.

8. If possible, it would be useful if the abstract contained a summary of the discrepancies in the cancer diagnoses dates as this is an essential consideration in cancer epidemiology studies.

Author response: We have added a brief comment about discrepancies in cancer diagnosis dates to the abstract.

Author change:

Before: (p2, line 41) Sensitivities were generally lower when primary care or hospitalisation data were used alone.

After: (p2, line 40) When primary care or hospitalisation data were used alone, sensitivities were generally lower and diagnosis dates were delayed.

9. Throughout the manuscript, the authors could be more specific about which set of combinations of datasets they are referring to as it was difficult to follow at times.

Author response: We have reviewed the manuscript and clarified which set of combinations of datasets we are referring to where this was unclear.

10. The conclusions should include the caveat that the ascertainment of incident cases using a combination of primary care, hospitalisation and death registration data was suboptimal for some cancer sites.

Author response: We have further emphasised that using a combination of CPRD GOLD, HES APC and ONS mortality is good for the majority of cancers but not all.

Author change:

Before: (p2, line 44) Ascertainment of incident cancers was good when using cancer registration data alone or in combination with other datasets, and for the majority of cancers when using a combination of primary care, hospitalisation and death registration data.

After: (p2, line 45) Ascertainment of incident cancers was good when using cancer registration data alone or in combination with other datasets, and for the majority of cancers when using a combination of primary care, hospitalisation and death registration data.

Before: (p13, line 304) Completeness and accuracy of recording of cancers in English data sources is high particularly when using NCRAS cancer registration data alone or in any combination with other data sources, and when using a combination of CPRD GOLD, HES APC and ONS mortality data, with variation between cancer types.

After: (p14, line 336) Completeness and accuracy of recording of cancers in English data sources is high particularly when using NCRAS cancer registration data alone or in any combination with other data sources, and for the majority of cancers when using a combination of CPRD GOLD, HES APC and ONS mortality data.

11. The abstract and methods should specify more clearly that this analysis is restricted to datasets within England and are not applicable to the UK as a whole.

Author response: We have further emphasized that our study is restricted to England in the abstract and methodology.

Author change:

Before: (p2, line 23) Our descriptive study uses linked Clinical Practice Research Datalink primary care; cancer registration; hospitalisation and death registration data.

After: (p2, line 24) Our descriptive study uses linked English Clinical Practice Research Datalink primary care; cancer registration; hospitalisation and death registration data.

Before: (p5, line 95) We completed a concordance study using linked² CPRD GOLD, HES APC, ONS mortality and NCRAS data.

After: (p5, line 98) We completed a concordance study using linked² English CPRD GOLD, HES APC, ONS mortality and NCRAS data.

12. Line 170 – what does ‘set 13’ refer to?

Author response: CPRD regularly updates the linkages to include newly participating practices and registered patients and extended data coverage periods for the linked data. Each release gets a new set number.

Author change:

Before: (p5, line 95). We completed a concordance study using linked CPRD GOLD, HES APC, ONS mortality and NCRAS data (January 2017 CPRD build, set 13 linkage data, study period 1 Jan 2000 – 31 December 2014).

After: (p5, line 95). We completed a concordance study using linked² English CPRD GOLD, HES APC, ONS mortality and NCRAS data. CPRD GOLD data were extracted from the January 2017 monthly release and the 13th update to CPRD's linked data. The study period was 1 Jan 2000 – 31 December 2014, with 31 December matching the end of the NCRAS coverage period.

VERSION 2 – REVIEW

REVIEWER	David Goldsbury Cancer Council NSW, Australia
REVIEW RETURNED	14-May-2020
GENERAL COMMENTS	The authors are to be commended on their clear and effective job of addressing the comments and revising the paper. I have only one very minor change request: in the abstract, results for completeness have been added, they just need some punctuation in “minimum 76.0% 2012 86.4% 2014 ...”. It is also worth noting that the response about improvement in staging data was very informative. In the manuscript this information is in a reference rather than being explicitly stated – if space allowed it might make a useful inclusion in the manuscript, but it is OK as is. Similarly, more explicit statements about the level of coverage of included GPs or the ease of data access would be useful. However these are not issues that should prevent publication.